# Severe postpartum haemorrhage at a large referral hospital in Uganda: A prospective observational pilot study

**Mia Appelbäck**[1,2]*, **Clare Lubulwa**[3], **Lawrence Kazibwe**[4], **Knut Haakon Stensæth**[5,6], **Thorkild Tylleskär**[1©], **Josaphat Byamugisha**[7©]

1 Centre for International Health, Department of Global Public Health and Primary Care, University of Bergen, Bergen, Norway, 2 Department of Obstetrics and Gynecology, Skåne University Hospital, Malmö, Lund, Sweden, 3 Mulago Specialised Women and Neonatal Hospital, Kampala, Uganda, 4 Kawempe National Referral Hospital, Kampala, Uganda, 5 Department of Radiology and Nuclear Medicine, St Olav's University Hospital, Trondheim, Norway, 6 Department of Circulation and Medical Imaging, Norwegian University of Science and Technology, Trondheim, Norway, 7 Department of Obstetrics & Gynaecology, College of Health Sciences, Makerere University, Kampala, Uganda

© These authors contributed equally to this work.

* mia.appelback@uib.no

## Abstract

### Background

Postpartum haemorrhage (PPH) is a leading cause of maternal mortality, near-misses and morbidity in Uganda and globally. Kawempe National Referral Hospital (KNRH), Uganda's largest obstetric referral hospital, receives many obstetric emergencies, including PPH, from lower-level health facilities. Little is known about the outcomes and management of severe PPH at KNRH. This study aimed to map the occurrence, profile and management challenges of severe PPH at KNRH.

### Methods

A prospective hospital-based observational pilot study was conducted between 5th April and 30th May 2023 at KNRH. Sixty women with severe PPH, both in-house and referrals, were enrolled. Data collection was done by research assistants on day 0–3 and 42 of inclusion, capturing characteristics, management and outcomes of the participants. Descriptive statistics were used for analysis.

### Results

Of the 60 participants, 47 were referrals. There were 3 maternal deaths, 56 maternal near-misses (hysterectomy, cardiovascular dysfunction, uterine rupture or massive blood loss, and 46 underwent critical interventions (intensive care, laparotomy or blood transfusions). All the participants with uterine ruptures and 12 out of the 13 participants with stillbirths and hysterectomies, respectively, were among referrals. Only

**Data availability statement:** The data that support the findings of this study are not openly available due to reasons of sensitivity and are available from the corresponding author upon reasonable request. Data are located in controlled access data storage at the University of Bergen. The local ethics committee can be contacted on rek-vest@uib.no (https://rekportalen.no/#omrek/REK_vest).

**Funding:** MAs PhD is financed through a grant from the Research Council of Norway (RCN), Norway, FRIPRO grant 324933. https://www.forskningsradet.no/en/. The funders had no role in study design, data collection and analysis, decision to publish, or preparation of the manuscript.

**Competing interests:** The authors have declared that no competing interests exist.

referrals had an initial systolic blood pressure ≤ 60 mm Hg and received ≥ 5 units of blood. In both groups approximately two thirds received tranexamic acid and oxytocin/misoprostol while few of the participants (23.3%) received uterine massage. Nine women had surgical site infections (8 were referrals), and 15 had suboptimal (fair/poor) wellbeing at 6 weeks postpartum (13 were referrals). None of the comparisons were statistically significant due to too few observations.

## Conclusions

Referrals were more critically ill and disproportionally affected by adverse outcomes and substandard care. While drugs and fluids were often timely administered according to national guidelines and did not differ greatly between the groups, gaps remained particularly non-pharmaceutical interventions. Bettered implementation of evidence-based PPH management and strengthening of the referral system could improve quality of care and maternal outcomes.

---

## Introduction

The global maternal mortality ratio (MMR) decreased by over 40% from 2000 to 2023 [1]. Yet, an estimated 260 000 women still die each year, with postpartum haemorrhage (PPH) as the leading direct cause [1,2]. Women in low-income countries have an increased risk of severe PPH, which not only contributes to maternal mortality but also to maternal near-misses (nearly died but survived) and maternal morbidity globally [3–5].

In Uganda, according to Demographic and Health Surveys (DHS) released in 2022, the MMR was 189/100,000 live births, with PPH being the leading cause, contributing to approximately 35% of all maternal deaths and significantly to maternal near-misses [6–8]. Kawempe National Referral Hospital (KNRH) is the largest women's national referral hospital in Uganda and receives many obstetric emergencies, including PPH, referred from lower-level health facilities. PPH has been reported as one of the leading cause of maternal death and maternal near-misses among intrapartum referrals to the national referral level in Uganda, yet there is no research on the profile and occurrence of patients presenting with severe PPH at KNRH [9]. Although maternal death due to severe PPH is common at lower-level facilities, institutional MMR is highest at referral and large private not-for-profit hospitals, where 50% of all maternal deaths occur, indicating the need to improve the referral system and quality of care given within the healthcare system [7].

The 'third delay' in the 3-delay model – receiving appropriate care at healthcare facilities – is a major contributor to maternal morbidity and mortality globally [10,11]. Shortage of supplies, of blood and of staff are common causes of the third delay in low- and middle-income countries (LMICs) [12]. In Uganda, nearly 50% of all avoidable factors associated with maternal death can be classified as delays in providing appropriate care at health facilities [7]. Challenges in providing responsive emergency obstetric and neonatal care (EmONC) in LMICs include delays in receiving

care and referrals, where lack of skills is a contributing factor [12]. PPH requires timely diagnosis and management and appropriate treatment [13]. A study from Uganda and Ghana showed overall good structural readiness in providing PPH care, particularly in the provision of uterotonics, but with gaps in the provision of tranexamic acid [14]. The WHO recommends formal protocols for the prevention and treatment of PPH and for patient referral, which Uganda has implemented [15,16]. Studies have shown that adherence to evidence-based guidelines is low in many LMICs, with providers fulfilling only 64% of recommended care for labour and delivery [11]. In lower-level facilities in Uganda there is a gap between knowledge and quality, where quality is measured by what essential care actions were observed in practice [17]. Less is known about how essential care actions are followed in higher-level facilities in Uganda.

This observational study aimed to provide information on how national guidelines were followed and to shed light on the outcomes and morbidities associated with severe PPH at KNRH for both in-house and referred patients. With about 90% of all births in Uganda occurring at facilities, it is paramount that the care received is of good quality if maternal mortality and morbidity are to be reduced [6]. Also, we intended to map the occurrence and profile of severe PPH as well as the current challenges faced in its management at Uganda's largest obstetric referral hospital.

## Methods

### Study design

This study was a prospective hospital-based observational study of 60 consecutive women with an on-going severe PPH. It was a pilot study for a randomised control trial. Data collection was done by research assistants (RAs) on day 0–3 and day 42 of inclusion.

### Study setting

In Uganda, the public healthcare system is divided into village health teams (VHTs), Health Centres (HC) II, III and IV, General Hospitals (GH), Regional Referral Hospitals (RRH), National Referral Hospitals (NRH) and specialised hospitals. Basic emergency obstetric and neonatal care (EmONC) should be available at HC IIIs, higher and Comprehensive EmONC at HC IVs and above [18]. The study took place at Kawempe National Referral Hospital (KNRH) in Kampala, Uganda, including all units (admissions, delivery, wards and operating theatres). The hospital has approximately 22 000 deliveries annually and is a referral hospital for many urban clinics, health centres and hospitals [19]. Recruitment took place from 5th April to 30th May 2023 and during this time frame 3409 deliveries took place at KNRH. With 13 of the 60 participants being in-house patients, severe PPH occurred in 0.004% of the deliveries.

### Data collection tools and procedures

The study included women who had delivered and developed severe PPH. The inclusion criteria were women 18 years and older and emancipated minors (15–17 years of age) with PPH and a systolic blood pressure of 90 mm Hg or below. Recruitment was done consecutively by RAs who were stationed at KNRH around the clock for the 2 months during the recruitment period.

Women with PPH were screened for eligibility and if fulfilled, informed written consent was obtained from the woman or her next-of-kin depending on the current condition of the woman. Research assistants did not participate in the management of PPH, but registered time of events, recorded when actions were taken and obtained Hb concentrations. The management of PPH with the different measures taken and the timeline from diagnosis to when the bleeding was stopped was documented, as well as the participants' vital parameters and Hb concentration. Hb was analysed using the point-of-care HemoCue Hb 301 (Ängelholm, Sweden). Once the participant was stable on day 0 or 1 of inclusion in the study, socio-demographic and clinical characteristics of the participant was recorded, including previous obstetric history and information on the current pregnancy and delivery. Vital parameters, Hb concentration and other relevant information on

management and outcomes was captured on days 1, 2 and 3. The final data collection was done 42 days after the initial event in a follow-up consultation with the RAs, where Hb concentration, general well-being and any complications were recorded. Participants unable to participate in person in this final follow-up were interviewed by telephone.

All data were captured on paper and doubly entered in EpiData 3.1 (http://www.epidata.dk), verified and merged into one dataset.

Initial quality control of the CRFs was done by a RA, then two senior researchers (MA and CL) checked the data for completeness and correctness.

## Variables

The operational definition of severe PPH was: 1) on-going bleeding and 2) a systolic blood pressure of 90 mm Hg or below. The primary outcomes were maternal death and/or emergency hysterectomy. Additional variables collected included initial blood pressure and pulse, shock index, Hb concentrations, and management of PPH such as uterotonic drugs given, uterine massage, blood transfusions and additional haemostatic interventions. Shock index (pulse in beats per minute (BPM) divided by systolic blood pressure, SBP) above 1.0 was used as a sign of shock [16].

ICU admission, laparotomy (hysterectomy and/or B-lynch sutures) and blood transfusions are examples of critical interventions recorded [5]. The data collected allowed the assessment of cardiovascular dysfunction (e.g., shock and continuous use of vasoactive drugs), coagulation dysfunction (e.g., massive blood transfusion) and uterine dysfunction (e.g., hysterectomy due to uterine haemorrhage), all criteria for life-threatening conditions according to the WHO near-miss approach [5]. These organ system-based criteria were used to define a maternal near-miss in this study. With this approach, shock was defined as a persistent systolic blood pressure of 80 mm Hg or below and/or 90 mm Hg and below with a pulse or 120 beats per minute (bpm) and above [5]. Continuous administration of vasoactive drugs (a WHO criterion for near-miss) was not always practiced in this setting, depending on the provider and availability of drip count equipment or drugs. Instead, we considered this criterion fulfilled if at least two vasoactive drugs or injections were administered. In this study, we used the sub-Saharan African adaption of the WHO maternal-near miss criteria specifically for defining 'massive blood transfusions' as 2 or more units of red blood cells (instead of 5) and recognising 'uterine rupture' as a criterion on its own [20].

Variables reflecting quality healthcare include 'prolonged labour' (longer than 24 hours from time of onset of labour to delivery), 'delay in blood transfusion', 'surgical site infection' and 'reported wellbeing 6 weeks postpartum' [11,21,22]. PPH can result in death on average 2 hours from onset if treatment is not initiated and thus 2 hours was used as the cut-off for delay in blood transfusion [23]. 'Reported wellbeing 6 weeks postpartum' was categorised as being good or suboptimal (fair or poor). Fair was paired with poor as many that had labelled their wellbeing as fair described a very challenging situation and appeared obviously affected by it, and yet chose fair over poor. Together these were coined suboptimal wellbeing in comparision to good wellbeing. Other indicators of suboptimal care were moderate to severe anaemia prior to arrival and long decision to incision times for Caesarean section (CS). WHO defines severe anaemia during pregnancy as Hb below 7 g/dl and moderate as Hb 7–9.9 g/dl and anaemia prior to delivery reflect suboptimal antenatal care [24]. Decision-incision time was calculated from the time of decision for CS to the time the CS started. It is recommended that the decision to delivery time for category 1 (immediate threat to the life of the woman or foetus) CSs be within 30 minutes to reduce the risk of neonatal complications and within 75 minutes for category 2 CSs (maternal or foetal compromise). Thus, long decision to incision time was defined as more than 75 minutes [25].

Variables on management of PPH were selected from Uganda's national guidelines on the management of PPH. The first response bundle in the guidelines includes a) uterine massage, b) uterotonics (oxytocin 10 international units (IU) intravenously or misoprostol 800 μg or ergometrine 0.2 mg), c) administration of 2 litres of fluid, d) bladder emptied and e) 1g tranexamic acid given [16]. Misoprostol was administered rectally and/or sublingually. Most patients have an indwelling catheter in situ at time of inclusion in the study thus `bladder emptied´ was under-reported in data collection and not

included in the results. Ten minutes was used as a cut-off time for the administration of fluid and oxytocin as it appears that oxytocin within 10 minutes of diagnosis of PPH is associated with a decreased risk of severe PPH, and it is also the international recommendation for triage after arrival at a facility for obstetric emergencies [26,27].

### Study size

Recruitment was planned to take place during 3 months or when a total of 60 participants were recruited. The latter was reached approximately two months after the study start. A sample size was not calculated as the objective was to obtain descriptive data to customise a subsequent intervention with the aim of improving standard of care and reducing maternal morbidity and mortality because of severe PPH.

### Quantitative variables and statistical methods

The dataset was exported to Rstudio (R version 4.3.0, 2023-04-32 ucrt) for data analysis. Variables were presented descriptively and stratified into in-house and referrals to allow comparison. Due to too few observations none of the comparisons were statistically significant and only the number and procentage of the variables are presented in the results.

### Ethics approval and consent to participate

Approval for this study was received from the Institutional Review Board (IRB), Makerere School of Medicine (Mac-SOMREC-2022–7457) and Regional Ethics Committee for Medical Research Ethics South East Norway (522376). Informed written consent was obtained from participants and if illiterate, a thumb print with the signature of a witness was used. Approval for recruitment of 'emancipated minors' (minors that are pregnant are considered 'emancipated minors' and are legally allowed to consent themselves) was given and a separate, tailored consent form was given to obtain consent from the minor themselves. As the participants recruited were in a critical condition, the initial informed, written consent was gained from a next-of-kin, with this being approved by the local ethics committee.

### Inclusivity in global research

Additional information regarding the ethical, cultural, and scientific considerations specific to inclusivity in global research is included in the Supporting Information (S1 Checklist)

### STROBE

This manuscript was prepared according to the STROBE guidelines for observational studies.

## Results

### Demographic and clinical characteristics

Of the 60 participants, 47 where referrals and 13 were in-house patients (Table 1). There were no non-responses. Of the 60 participants, 37 were delivered by CS and of these, 30 were referrals. A third of in-house patients and one sixth of referrals were primigravida.

### Outcomes

All the uterine ruptures occurred in the referral group, and 25.5% of referrals compared to 7.7% of in-house patients underwent hysterectomy (Table 2). Despite this, the proportion of patients that fulfilled the criteria for maternal near-miss and received critical interventions was similar in the two groups. Of the 13 stillbirths in the study, 12 were referrals and 1 was in-house.

**Table 1. Characteristics of the 60 study participants with severe PPH by referral status.**

| | In-house (N = 13) n (%) | Referrals (N = 47) n (%) | Overall (N = 60) n (%) |
|---|---|---|---|
| **Age (years)** | | | |
| 10-19 | 2 (15.4) | 2 (4.3) | 4 (6.7) |
| 20-29 | 2 (15.4) | 21 (44.7) | 23 (38.3) |
| 30-39 | 8 (61.5) | 21 (44.7) | 29 (48.3) |
| 40-49 | 1 (7.7) | 3 (6.4) | 4 (6.7) |
| **Education** | | | |
| None or incomplete primary | 0 (0) | 9 (19.1) | 9 (15.0) |
| Completed primary | 6 (46.2) | 13 (27.7) | 19 (31.7) |
| Completed secondary or higher | 7 (53.8) | 24 (51.1) | 31 (51.7) |
| Unknown | 0 (0) | 1 (2.1) | 1 (1.7) |
| **Marital status** | | | |
| Married/cohabiting | 12 (92.3) | 41 (87.2) | 53 (88.3) |
| Single/separated/widowed | 1 (7.7) | 6 (12.8) | 7 (11.7) |
| **Parity** | | | |
| Primigravida | 5 (38.5) | 7 (14.9) | 12 (20.0) |
| 1-3 | 4 (30.8) | 22 (46.8) | 26 (43.3) |
| 4+ | 4 (30.8) | 18 (38.3) | 22 (36.7) |
| **Mode of previous deliveries** | | | |
| All Caesarean sections (CS) | 1 (7.7) | 6 (12.8) | 7 (11.7) |
| All vaginal deliveries (VD) | 5 (38.5) | 25 (53.2) | 30 (50.0) |
| Both VDs and CSs | 2 (15.4) | 9 (19.1) | 11 (18.3) |
| No previous deliveries | 5 (38.5) | 7 (14.9) | 12 (20.0) |
| **Term or pre-term birth** | | | |
| Pre-term birth | 1 (7.7) | 12 (25.5) | 13 (21.7) |
| Term birth | 12 (92.3) | 35 (74.5) | 47 (78.3) |
| **Mode of current delivery prior to PPH** | | | |
| Caesarean section | 7 (53.8) | 30 (63.8) | 37 (61.7) |
| Vaginal delivery | 6 (46.2) | 17 (36.2) | 23 (38.3) |

**Table 2. Adverse outcomes in the study by referral status.**

| | In-house (N = 13) n (%) | Referral (N = 47) n (%) | Overall (N = 60) n (%) |
|---|---|---|---|
| Uterine rupture | 0 (0) | 9 (19.1) | 9 (15.0) |
| Cardiovascular dysfunction[a] | 10 (76.9) | 42 (89.4) | 52 (86.7) |
| Transfused ≥ 2 blood units | 9 (69.2) | 35 (74.5) | 44 (73.3) |
| Hysterectomy | 1 (7.7) | 12 (25.5) | 13 (21.7) |
| Maternal near-miss (≥ 1 of above) | 12 (92.3) | 44 (93.6) | 56 (93.3) |
| Critical intervention[b] | 10 (76.9) | 37 (78.7) | 47 (78.3) |
| Stillbirth | 1 (7.7) | 12 (25.5) | 13 (21.7) |
| Maternal death | 1 (7.7) | 2 (4.3) | 3 (5.0) |

[a]Shock or administration of vasoactive drugs (S1 Table).

[b]Laparotomy, admission to ICU or blood transfusion (S2 Table).

## Key clinical indicators

The initial systolic blood pressures in the study of between 41–60 mm Hg were recorded from 5 referred and 0 in-house patients (S1 Fig).

At diagnosis of PPH, in-house patients had a median SBP of 79 mm Hg and referrals of 83 mm Hg. It took in-house patients on average 10 minutes from diagnosis of PPH to reach a median SBP greater than 100 mm Hg compared to 30 minutes for referrals (Fig 1).

Referred patients had consistently a higher shock index compared to in-house patients the first hour after diagnosis of PPH, and only referrals received 5 or more units of blood (Fig 2 and S2 Fig.)

## Management

Approximately two thirds of the participants received tranexamic acid and the recommended oxytocin or misoprostol dosages (Table 3). None received ergometrine. Out of the 40 patients who received the recommended dosages, 33 received oxytocin and/or misoprostol within 10 minutes of diagnosis of PPH. Only 1 in 5 patients received

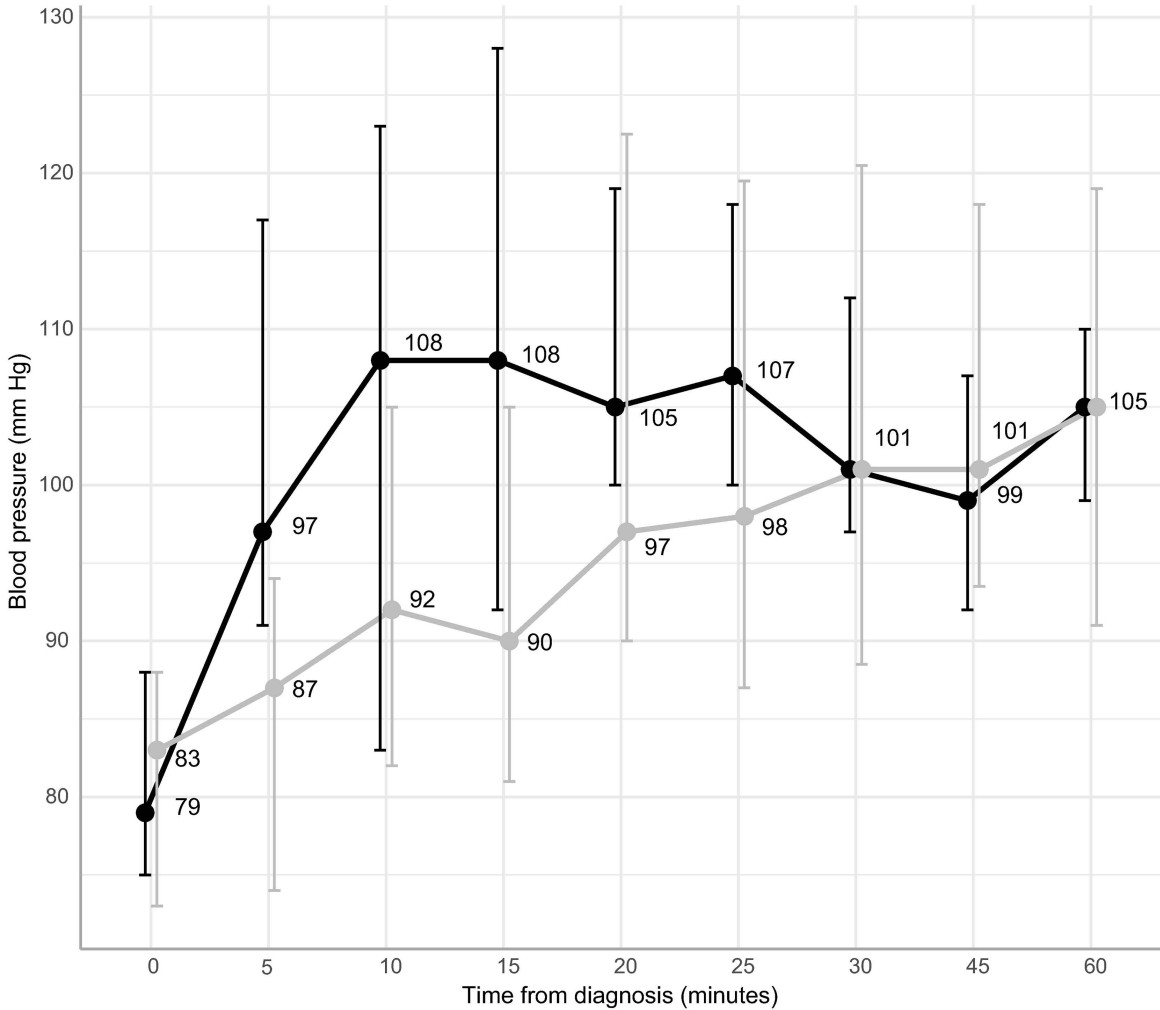

**Fig 1. The median (± interquartile range, IQR) systolic blood pressure (mm Hg) during the first hour from diagnosis of PPH, by referral status.**

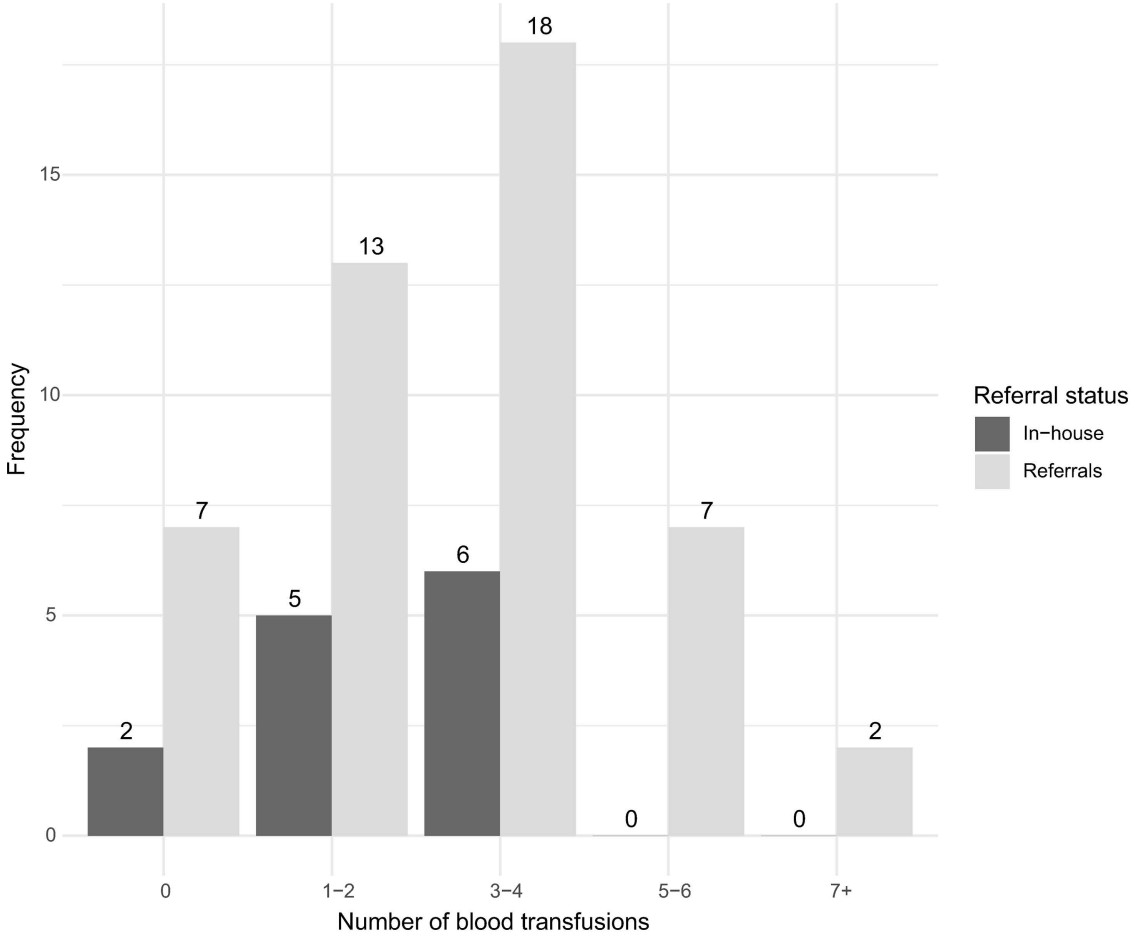

**Fig 2. Number of blood transfusions received, by referral status.**

**Table 3. First response bundle received by patients with severe PPH, by referral status.**

|  | In-house (N = 13) n (%) | Referral (N = 47) n (%) | Overall (N = 60) n (%) |
|---|---|---|---|
| Uterine massage | 4 (30.8) | 10 (21.3) | 14 (23.3) |
| Uterotonics (Oxytocin ≥ 10 IU or misoprostol ≥ 800 μg) | 9 (69.2) | 31 (66.0) | 40 (66.7) |
| Uterotonics given within 10 minutes | 7 (53.8) | 26 (55.3) | 33 (55.0) |
| Fluids ≥ 2 litres (Ringer or normal saline) | 5 (38.5) | 7 (14.9) | 12 (20.0) |
| Fluids given within 10 minutes | 11 (84.6) | 35 (74.5) | 46 (76.7) |
| Tranexamic acid ≥ 1g | 7 (53.8) | 30 (63.8) | 37 (61.7) |

uterine massage. Fifteen percent of referrals and 39% of in-house patients received 2 litres of fluid or more. Fluid management was initiated within 10 minutes of diagnosis for most patients. In cases where atony or retained products were causes of the PPH, 10 out of the 41 received Carboprost, and 19 out of the 41 received an additional oxytocin 20 IU (S3 Table).

## Quality indicators

Antenatal Hb was missing in more than half of the participants (Table 4). Moderate or severe anaemia prior to admission occurred only in referrals. Prolonged labour occurred in 1 of 5 participants. Of the 16 that had a decision-incision time longer than 75 minutes, 15 of these were referrals whereas blood transfusion delay occurred in 5 participants in each group. Surgical site infection occurred in 9 participants with 8 of these being referrals, who also accounted for 13 out of the 15 participants that reported suboptimal wellbeing postpartum.

All in-house patients had a decision-incision time within 2 hours with 3 referrals experiencing a decision-incision time between 4–8 hours (S3 Fig). Of those that had a labour lasting over 48 hours, 6 were referrals and 1 was in-house (S4 Fig).

## Discussion

The objective was to map the occurrence and profile of severe PPH as well as the current challenges faced in its management at KNRH. This observational study involved 60 participants with severe PPH over 2 months, mostly referred patients. Referrals were also overrepresented when it came to adverse outcomes such as uterine rupture, hysterectomy, and stillbirth, although maternal near-miss and exposure to critical interventions were similar between groups. Referrals were in a more

**Table 4. Quality indicators, by referral status.**

| | In-house (N = 13) n (%) | Referrals (N = 47) n (%) | Overall (N = 60) n (%) |
|---|---|---|---|
| **Haemoglobin levels prior to admission** | | | |
| Hb ≥ 10 g/dl | 5 (38.5) | 11 (23.4) | 16 (26.7) |
| Moderate anaemia (Hb 7–9.9 g/dl) | 0 (0) | 7 (14.9) | 7 (11.7) |
| Severe anaemia (Hb < 7 g/dl) | 0 (0) | 4 (8.5) | 4 (6.7) |
| Hb not recorded | 8 (61.5) | 25 (53.2) | 33 (55.0) |
| **Prolonged labour** | | | |
| Within 24 hours | 9 (69.2) | 28 (59.6) | 37 (61.7) |
| Longer than 24 hours | 3 (23.1) | 10 (21.3) | 13 (21.7) |
| Missing data | 1 (7.7) | 9 (19.1) | 10 (16.7) |
| **Decision-incision time for CS** | | | |
| Within 75 mins | 6 (46.2) | 14 (29.8) | 20 (33.3) |
| Longer than 75 mins | 1 (7.7) | 15 (31.9) | 16 (26.7) |
| Missing data/vaginal delivery | 6 (46.2) | 18 (38.3) | 24 (40.0) |
| **Blood transfusion** | | | |
| Within 2 hours | 5 (38.5) | 32 (68.1) | 37 (61.7) |
| Delayed (> 2 hours) | 5 (38.5) | 5 (10.6) | 10 (16.7) |
| Not transfused | 3 (23.1) | 10 (21.3) | 13 (21.7) |
| **Surgical site infection** | | | |
| No surgical site infection | 11 (84.6) | 37 (78.7) | 48 (80.0) |
| Surgical site infection | 1 (7.7) | 8 (17.0) | 9 (15.0) |
| Missing (maternal deaths) | 1 (7.7) | 2 (4.3) | 3 (5.0) |
| **Wellbeing 6 weeks postpartum** | | | |
| Good | 10 (76.9) | 32 (68.1) | 42 (70.0) |
| Suboptimal (fair or poor) | 2 (15.4) | 13 (27.7) | 15 (25.0) |
| Missing (maternal deaths) | 1 (7.7) | 2 (4.3) | 3 (5.0) |

critical condition on inclusion, with higher shock index and lower SBP, and requiring more blood transfusions and a longer time for SBP to reach 100 mm Hg. Management of PPH, including timely administration of drugs and fluids, was comparable for the two groups, although uterine massage was underutilised, and referrals were less likely to receive 2 litres or more of fluids. Indicators of subquality care, such as antepartum anaemia, surgical site infection, decision-incision delays, and postpartum wellbeing, were more common among referrals, while in-house patients experienced more blood transfusion delays.

### Referrals arriving in a critical condition

Referrals arrived in more critical condition, with higher SI index and lower SBPs on inclusion. Studies from Uganda suggest that referrals are arriving in a critical condition to referral hospitals and at risk of adverse outcomes following initial management and attempted delivery at lower-level health facilities [28,29]. Lack of knowledge, large know-do gaps at lower-level facilities especially for the prevention of PPH and the inability to provide comprehensive EmONC services, even at HC IVs where these services should be available, could contribute to this [17,28]. Other studies from LMICs have reported subquality care at lower-level facilities prior to referral and high-quality care being received at referral hospitals [30,31]. Gaps in services and quality of care at lower-level facilities could provide an explanation of why referrals were more likely to have adverse outcomes in our study, highlighting the need to strengthen lower-level facilities.

### Adverse outcomes: Referrals overrepresented

Other studies have shown that referrals are overrepresented when it comes to maternal near-miss in Uganda [32]. Although our study showed no great difference in maternal death or maternal near-miss between referrals and in-house patients, referrals were overrepresented in adverse outcomes like uterine rupture and hysterectomy. This is consistent with other studies showing increased risk of maternal morbidities and severe outcomes with reports of subquality care and delays in the referral system in many LMICs [33–35]. Being referred has been reported as a risk factor for stillbirth and uterine rupture in obstructed labour cases [36,37], which is also reflected in our findings.

### Management indicators and following guidelines

Timely and evidence-based management of PPH is paramount. Approximately two thirds of participants received uterotonics and tranexamic acid and one fifth at least 2 litres of fluids. Our findings align with other studies on oxytocin or uterotonics administration (45.5–93%) and fluid resuscitation (28%) [38–40]. Tranexamic acid provision was higher in our study, potentially illustrating a change in routines over time [14,41]. Most participants received fluid management and uterotonics within 10 minutes of diagnosis, which is higher than reported in one study [42]. Only 14.9% of referrals received 2 litres of fluid compared to 38.5% of in-house patients although the latter were more likely to experience blood transfusion delays. These two findings could potentially go hand-in-hand with referrals getting more blood and in-house patients getting more fluid. A study in Kenya showed high adherence to PPH management guidelines such as uterine massage in referral hospitals was done in 92% of cases [41]. Our study showed lower adherence to uterine massage which could possibly be explained by gaps in knowledge or skills, patient volumes, gaps in staff coverage and/or supply shortages, factor previously reported being barriers to receiving timely and evidence-based care [12,14,17,43,44]. Knowledge gap is less likely at the national referral level yet a study on know-do gaps in obstetric care at lower-level facilities in Uganda showed that lower quality scores are more common when patient volumes are high, which could offer a potential explanation to our findings [17]. Overall, drug and fluid administration were consistent with other studies, though simpler management strategies could be better implemented to improve PPH management without costly interventions.

### Quality of care in the healthcare system

Many quality-of-care indicators in our study signal areas of potential improvement, including optimising Hb management during pregnancy, preventing prolonged labour, and shortening decision-to-incision times, thus reflecting that quality

maternal healthcare is not limited to one level of the healthcare system. Common issues included undetected and unmanaged anaemia reflecting substandard management of anaemia during pregnancy, consistent previously repots of severe anaemia contributing to 18% of maternal deaths in 2022/2023 with only 12% of all the maternal deaths having registered Hb from ANC [7]. Prolonged labour, an indicator of poor-quality obstetric care, affected 20% of participants indicating systemic delays in seeking, reaching, or receiving care [10]. Delay in blood transfusions is not uncommon in many LMICs and have been reported in the Ugandan setting [12,14,43]. Most participants received blood transfusion within 2 hours of diagnosis of PPH indicating effective routines at the national referral level for the acquisition and administration of blood products. Timely access to CSs (decision-incision time of within 75 minutes) was achieved in 20 out of 36 cases but delays also occurred, especially among referrals. A study from a referral hospital in Uganda reported that 61.7% of patients experienced at least one delay in receiving surgical care with lack of space and lack of supplies or personnel being the most common causes [45]. Long decision-incision times are common in LMICs and could be contributing to adverse perinatal and maternal outcomes [12]. In Uganda, 33% of FSBs delivered by CS had a decision-incision time exceeding an hour indicating more is needed to be done to improve timely management [7]. All our results, not only those defined as quality indicators, reflect broader system challenges with uterine rupture, stillbirths, evidence-based care, functional referral systems and maternal-near miss, all being proxies for quality obstetric care [46–49]. Challenges in providing responsive obstetric care and health system readiness in LMICs include decision-incision times and functional referral systems, crucial for EmONC responsiveness [12,44]. The referral system is intrinsically related to the quality of obstetric services provided and many of our findings, such as hysterectomies to surgical-site infections, being overrepresented in referrals indicates healthcare system shortcomings. Substandard care can lead to adverse outcomes, including maternal death, near-miss, and long-term health consequences, the latter of which was more common in referrals (13/15 with suboptimal wellbeing) [11,21].

Being referred has been described as a journey of vulnerability, with many from Uganda and neighbouring countries reporting delays and poor experiences leading to adverse health and psychosocial outcomes [34,35]. The WHO's quality indicators for maternal and newborn healthcare include provision of care such as evidence-based management and functioning referral systems, experience of care, competent human resources and availability of essential physical resources [49]. Our study highlights these domains, showing that while evidence-based management at KNRH is largely fulfilled, improvements are needed, especially in understanding and addressing gaps in the referral system. Strengthening the referral system and improving implementation of evidence-based guidelines such as non-pharmaceutical interventions in PPH management should inform maternal health practice and policy. Our findings not only reflect quality of obstetric care in the healthcare system, but addressing the issues raised here would require a health systems approach. Strategies proposed to reduce the risk for stillbirths encompass improving quality of care within the healthcare system, such as the referral system, EmONC and ANC, which would in turn lead to improvements for many other of the outcomes highlighted in this study [50]. Our study supports the need to strengthen and streamline the referral system and quality of care received by patients with severe PPH [7].

### Strengths and limitations

A strength of the study is that data collection was prospective done by trained midwives, although the emergency setting may have led to incomplete recordings potentially explaining why uterine massage was so low.

The study's small sample size means small discrepancies can affect proportions and increased focus on PPH could have resulted in improved detection and management. Some data were self-reported by mothers thus subject to biases, such as the starting time of labour which is why the definition of prolonged labour with the longest time of 24 hours was chosen.

### Conclusion

Referred patients were initially more critically ill and overrepresented in adverse outcomes, with indicators of substandard care being more common among referrals. While the timely administration of drugs and fluids often adhered to national

guidelines and showed little difference between referrals and in-house patients, shortcomings were noted such as the first response intervention of uterine massage. To improve maternal outcomes and quality of care, there is a need for better implementation of evidence-based PPH management, and strengthening of lower-level facilities and the referral system for obstetric emergencies.

## Supporting information

**S1 Table. Occurrence of cardiovascular dysfunction by referral status.**
(DOCX)

**S2 Table. Criteria for critical interventions.**
(DOCX)

**S3 Table. Additional oxytocin 20 IU or Carboprost given if atony and/or retained products was the cause of the PPH.**
(DOCX)

**S4 Table. Additional risk factors of pregnancy.**
(DOCX)

**S5 Table. Complications in labour.**
(DOCX)

**S1 Fig. Initial systolic blood pressure (mm Hg) by referral status.**
(TIF)

**S2 Fig. Median and interquartile range of shock index in the 1st hour after diagnosis of PPH.**
(TIF)

**S3 Fig. The decision-incision time for Caesarean sections, by referral status.**
(TIF)

**S4 Fig. Histogram showing time in labour by referral status.**
(TIF)

**S5 Fig. Median (± IQR) haemoglobin levels (g/dl) with interquartile range from prior to arrival to hospital to 42 days postpartum, by referral status.**
(TIF)

**S6 Fig. Histogram showing the causes of PPH by referral status (participants can have multiple causes).** Examples of other causes include placental abruption, placenta accreta spectrum, uterine tears and coagulopathies.
(TIF)

**S1 Checklist. Inclusivity in global research.**
(DOCX)

## Acknowledgments

We thank the participants of the study for their participation and the research assistants, the data manager Damien Wasswa and the project manager Hadija Nalubwama for their efforts in the study.

## Author contributions

**Conceptualization:** Clare Lubulwa, Lawrence Kazibwe, Knut Haakon Stensæth, Thorkild Tylleskär, Josaphat Byamugisha.

**Formal analysis:** Mia Appelbäck.

**Funding acquisition:** Thorkild Tylleskär.

**Methodology:** Clare Lubulwa, Lawrence Kazibwe, Knut Haakon Stensæth, Thorkild Tylleskär, Josaphat Byamugisha.

**Project administration:** Clare Lubulwa, Lawrence Kazibwe, Knut Haakon Stensæth, Josaphat Byamugisha.

**Supervision:** Thorkild Tylleskär, Josaphat Byamugisha.

**Visualization:** Mia Appelbäck, Thorkild Tylleskär.

**Writing – original draft:** Mia Appelbäck, Thorkild Tylleskär.

**Writing – review & editing:** Clare Lubulwa, Lawrence Kazibwe, Knut Haakon Stensæth.

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
