## [Decision Letter · Decision Letter 0]

28 Apr 2025

Dear Dr. Appelbäck,

Thank you for submitting your manuscript to PLOS ONE. After careful consideration, we feel that it has merit but does not fully meet PLOS ONE’s publication criteria as it currently stands. Therefore, we invite you to submit a revised version of the manuscript that addresses the points raised during the review process.

We look forward to receiving your revised manuscript.

Kind regards,

Hale Teka

Academic Editor

PLOS ONE

Journal Requirements:

3. In this instance it seems there may be acceptable restrictions in place that prevent the public sharing of your minimal data. However, in line with our goal of ensuring long-term data availability to all interested researchers, PLOS’ Data Policy states that authors cannot be the sole named individuals responsible for ensuring data access (http://journals.plos.org/plosone/s/data-availability#loc-acceptable-data-sharing-methods).

Additional Editor Comments:

Dear Authors,

I commend you on a relevant maternal health issue. I kindly ask you to revise few minor comments.

Abstract

Result - "All the uterine ruptures and 12 out of 13 both stillbirths and hysterectomies were among referrals." Please rewrite this. This is not clear.

The results section fails to show level of significance. This has to be revised as referrals vs non-referrals with 95% CI and p-value.

Suboptimal wellbeing is not a medical term. What do you mean by suboptimal wellbeing?

Introduction - The data you used for the global maternal mortality is old. Please use recent reports by the interagency group released in 2025.

Methods - The justification put forth for the sample size is not scientific. It does not matter if your study is descriptive or analytic an appropriate sample size should be sought.

Results - Could you please calculate the p-value for each variable being compared in between the two groups to see the level of significance of the differences?

Discussion - please remove 'summary'

First paragraph - start with restating your objective, and then follow summarizing the main findings that you will subsequently discuss.

Most of the graphs are not necessary and they can be described within text.

Reviewers' comments:

Reviewer's Responses to Questions

**Comments to the Author**

1. Is the manuscript technically sound, and do the data support the conclusions?

Reviewer #1: Yes

Reviewer #2: Yes

2. Has the statistical analysis been performed appropriately and rigorously?

Reviewer #1: Yes

Reviewer #2: Yes

3. Have the authors made all data underlying the findings in their manuscript fully available?

Reviewer #1: Yes

Reviewer #2: Yes

4. Is the manuscript presented in an intelligible fashion and written in standard English?

Reviewer #1: Yes

Reviewer #2: Yes

Reviewer #1: Review of the Manuscript on Postpartum Hemorrhage Management at Kawempe National Referral Hospital

This manuscript addresses a critical public health issue — postpartum hemorrhage (PPH), which remains the leading cause of maternal mortality globally, particularly in low-income countries. The authors effectively highlight the significance of PPH as a major contributor to maternal deaths and severe maternal morbidity, particularly in Uganda, where PPH is responsible for approximately 35% of maternal deaths. The focus on Kawempe National Referral Hospital (KNRH) — Uganda’s largest obstetric referral hospital — is highly relevant, as this facility receives numerous emergency cases, including severe PPH. By investigating adherence to national guidelines, the study provides valuable insights into the quality of care delivered in this high-risk setting.

The study is well-prepared and presents significant findings that have the potential to inform both clinical practice and policy improvements. However, I suggest minor revisions to enhance clarity and ensure the manuscript’s completeness.

Suggested Revisions and Clarifications

Clarification of Acronym (DHS)

In line 6, the abbreviation DHS is mentioned for the first time without explanation. To ensure the readership understands its meaning, I recommend defining this acronym in its first use (e.g., Demographic and Health Survey).

Route of Misoprostol Administration

The manuscript references the use of misoprostol, but it is unclear which route of administration was employed — buccal, vaginal, or rectal. Given that the route of administration may significantly influence the drug’s efficacy and absorption rate, I suggest providing this detail.

Overall Assessment

This is a well-prepared manuscript that addresses an important and timely topic. The authors provide valuable insights into the challenges associated with managing severe PPH at KNRH, with a particular focus on adherence to national guidelines. The findings have important implications for improving the quality of obstetric care and reducing maternal morbidity and mortality in Uganda and other low-resource settings.

With the above minor revisions and clarifications, this manuscript will strongly contribute to the literature on maternal health. I commend the authors for their thorough work and thoughtful analysis.

Recommendation: Minor revisions are required.

Sincerely,

Reviewer #2: Severe postpartum haemorrhage at a large referral hospital in Uganda: a prospective observational study

Comments

1. Abstract

• Background: Does not provide the rationale for the study

• Methods: should clearly show the study period and summary of the analysis

2. Introduction

• Should contain the global MMR of the post-SDG era (decreeing, increasing, or stagnated?)

3. Methods

• Participants and procedure:

If the recruitment period was from 5th April to 30th May, approximately two months, the data collection period should be more than two months. Because the mother who was recruited on 30th May will be studied again after 42 days. Therefore, the data collection time is more than two months.

Why were the minors excluded from the study?

• Data collection tools and procedures

Instead of “participants and procedures”, I would recommend “data collection tools and procedures.”

Additionally, they have to show in this section that they used EpiData during data collection

• Sample size

The justification for the sample size to be 60 is not satisfactory.

• Data quality assurance

There is no evidence of efforts to ensure data quality

• The statistical methods:

Lacks specific details. What type of descriptive statistics have they used?

4. Results

• Should describe if there were non-responses and why.

• Remove the name of the figures in the body of the manuscript, e.g., on line 249, 263 and so on

5. Discussion

Remove the word “summary” on line 301

Implications of the study on maternal and child health practice and policy should be boldy discussed

**Do you want your identity to be public for this peer review?** For information about this choice, including consent withdrawal, please see our Privacy Policy

Reviewer #1: No

Reviewer #2: **Yes: ** Mengistu Hagazi Tequare

---

## [Author Response · Author response to Decision Letter 1]

1 Jul 2025

1. Format changes have been made according to PLOS ONE guidelines

2. An inclusivity checklist for global research has been included

3. Contact information to the ethics committee for access to data has been included

4. The reference list has been double checked

---

## [Editor Report · Decision Letter 1]

16 Jul 2025

PLOS ONE

Dear Dr. Mia Appelbäck,

Thank you for submitting your manuscript to PLOS ONE. After careful consideration, we feel that it has merit but does not fully meet PLOS ONE’s publication criteria as it currently stands. Therefore, we invite you to submit a revised version of the manuscript that addresses the points raised during the review process.

Please address the following few comments and resubmit the manuscript including two forms. 

1) Please state that this is a pilot study in the title, in the abstract, and the main body and state that a large RCT will be conducted. 

2) Please use option A for the resubmission. 

We look forward to receiving your revised manuscript.

Kind regards,

Hale Teka

Academic Editor

PLOS ONE

Journal Requirements:

Additional Editor Comments:

Dear Authors,

Thank you very much for addressing all my comments and the reviewers comments including preparing the revision in 2 options. 

Please address the following few comments and resubmit the manuscript including two forms.

1) Please state that this is a pilot study in the title, in the abstract, and the main body and state that a large RCT will be conducted.

2) Please use option A for the resubmission.

---

## [Author Response · Author response to Decision Letter 2]

13 Aug 2025

These have been addressed in the attached rebuttal letter.

---

## [Editor Report · Decision Letter 2]

18 Aug 2025

Severe postpartum haemorrhage at a large referral hospital in Uganda: a prospective observational pilot study

PONE-D-25-07686R2

Dear Mrs Mia Appelbäc, 

We’re pleased to inform you that your manuscript has been judged scientifically suitable for publication and will be formally accepted for publication once it meets all outstanding technical requirements.

Kind regards,

Hale Teka

Academic Editor

PLOS ONE
---

## [Editor Report · Acceptance letter]

PONE-D-25-07686R2

PLOS ONE

Dear Dr. Appelbäck,

I'm pleased to inform you that your manuscript has been deemed suitable for publication in PLOS ONE. Congratulations! Your manuscript is now being handed over to our production team.

Kind regards,

on behalf of

Dr. Hale Teka

Academic Editor

PLOS ONE